# Influence of Different Types of Small Hydropower Stations on Macroinvertebrate Communities in the Changjiang River Basin, China

**Weihua Zhao** [1,2,†], **Weijie Guo** [1,2,†], **Liangyuan Zhao** [1,2], **Qingyun Li** [1,2], **Xiaohuan Cao** [1,2] and **Xianqiang Tang** [1,2,*]

1   Basin Water Environmental Research Department, Changjiang River Scientific Research Institute, Wuhan 430010, China
2   Hubei Provincial Key Laboratory of River Basin Water Resources and Eco-Environmental Sciences, Changjiang River Scientific Research Institute, Wuhan 430010, China
*   Correspondence: ckyshj@126.com; Tel.: +86-27-8292-6192
†   The first two authors contributed equally to this work.

**Abstract:** Many studies have investigated the influence of hydropower stations on macroinvertebrate communities, but few have clarified the influence of different types of hydropower stations. A total of 133 samples obtained from seven rivers, on which 45 hydropower stations are located, with the rivers distributed across four provinces (Yunnan, Jiangxi, Fujian, and Hubei) were investigated to study the influence of different types of small hydropower stations on macroinvertebrate communities. Samples were collected during 2011–2012. Results showed that 126 taxa of macroinvertebrates were collected, of which 68.3% were insects. The average macroinvertebrate density and biomass were $966 \pm 112$ ind/m$^2$ and $17.31 \pm 1.54$ g/m$^2$, respectively. For dam-type hydropower stations, the intercepting effect of the dam was the main factor affecting macroinvertebrate populations, whereas the influence of hydrological period was nonsignificant. Macroinvertebrate taxa richness exhibited a gradual increase from reservoir reaches to down-dam reaches and then to natural reaches (4.4, 6.5, and 9.5, respectively). The Shannon–Wiener index showed a similar increasing trend (1.06, 1.48, and 1.58, respectively), whereas biomass levels exhibited a decreasing trend (56.3, 25.2, and 6.0 g/m$^2$, respectively). For the diversion-type hydropower stations, hydrological period was the main influential factor, whereas the intercepting effect of the dam was nonsignificant. From wet to dry seasons, increases were observed in macroinvertebrate abundance (5.2 to 8.3), density (322.2 to 1170.5 ind/m$^2$), biomass (24.6 to 40.1 g/m$^2$), and Shannon–Wiener index (1.23 to 2.08).

**Keywords:** small hydropower station; dam-type hydropower stations; diversion-type hydropower stations; macroinvertebrate communities; Changjiang River Basin

---

## 1. Introduction

As of the end of 2015, China has constructed more than 47,000 small hydropower stations with a capacity of less than 50,000 kW [1]. These small hydropower stations can be divided into two types (i.e., dam-type and diversion-type) according to the layout mode of the dam. Dam-type power stations are typically composed of a dam (specifically, a large dam that intercepts water and forms a reservoir upstream of dam) and a power plant, whereas diversion-type power stations are usually composed of a dam (small dam to raise water level and usually no significant reservoir is formed), a diversion channel, and a power plant. Most Chinese hydropower stations are located in the mountainous rivers of southern China because of the abundant water resources in these rivers. Previous studies have shown that changes in river morphology (e.g., riffles and rapid flow) and hydraulic conditions

(e.g., velocity and water depth) influence the benthic community structure [2–4]. In addition, the effect of external disturbances in small-scale water ecosystems is particularly evident, and the degradation rate of small watershed systems is faster, but the recovery time is relatively short [5,6].

Macroinvertebrates are one of the most widely distributed taxa of river habitats, and they are also a critical part of the ecological systems of rivers, playing a crucial role in material circulation and energy flow [7]. Macroinvertebrates are often utilized as an indicator to evaluate the changes in a water environment [7–9]. Although many studies have investigated the impact of hydropower stations on macroinvertebrates [10–17], their results have been inconsistent. Different types of hydropower stations have different interception effects as well as ecological effects [18]. Santucci et al. [19] investigated the effect of low dams on aquatic organisms, habitats, and water quality in a 171 km reach in the United States. Premstaller et al. used the fish and macroinvertebrates to quantify the effects of big reservoirs in Italy and the effects of hydropeaking downstream of the dam [20]. The results showed that water storage areas accounted for 55% of the total surface area of the river, and the macroinvertebrate indicators in the natural flow area were significantly higher than that in the storage area. Bredenhand and Samways [21] investigated the Tinau River in Nepal and showed that a small hydropower station had only a minor effect on the macroinvertebrate community structure. Cortes [22] studied the influence of small dams on macroinvertebrates in an unpolluted stream source and found that the effect of artificial regulation on flow appeared to be small and seemed to exert no obvious change on the physical habitat or water quality. Conclusions have thus been inconsistent regarding the influence of small hydropower stations on macroinvertebrates. It was shown that macroinvertebrates in small rivers exhibit a high degree of adaptability, which prevents researchers from identifying a clear biological response [23], and therefore it is difficult to reveal the causal relationship about the influence of small hydropower stations on macroinvertebrates.

In summary, most related studies have concentrated on the effect of hydropower stations on hydrological changes and macroinvertebrate community structures. Few studies have focused on the influence of different types of small hydropower stations [3,4]. Many small hydropower stations have been established in China, yet it is unclear whether they have had a significant influence on the macroinvertebrate community structure and whether the influence differs between different types of hydropower station (diversion-type and dam-type). Based on these research problems, researchers selected seven rivers in the Changjiang River Basin of China to investigate the influence of small hydropower stations on macroinvertebrates in this study.

## 2. Research Area and Methods

### 2.1. Research Area

During the period 2011–2012, collections of invertebrates were made and habitat parameters were monitored in the seven investigated rivers. The seven rivers were chosen in that they are distributed in different ecological hydrological divisions in China, where numerous dam-type hydropower stations and diversion-type hydropower stations are distributed. The sample sites are shown in Figure 1. Considering that changes in the habitat elements in different hydrological periods may have different effects on the biological community, parts of the rivers were investigated in both the wet and dry seasons. According to the magnitude of discharge and the water level, from the upper to the lower reach, the river could be divided into five different sections (reservoir reaches, down-dam reaches, dewatered reaches, recovered-water reaches and natural reaches) (Figure 2, using diversion-type power stations as an example). These five parts were the investigated parts in the study. Each standard sample of biological quality consisted of 2–3 replications.

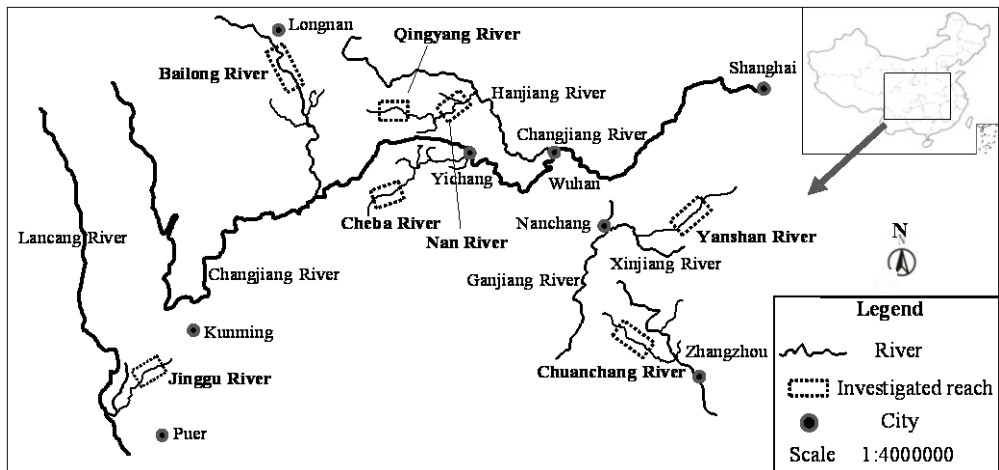

**Figure 1.** Schematic map of seven investigated rivers in the study (The scale is only for the Changjiang River and Lancang River. Scales of seven investigated rivers have been artificially enlarged to show their location and shape clearly).

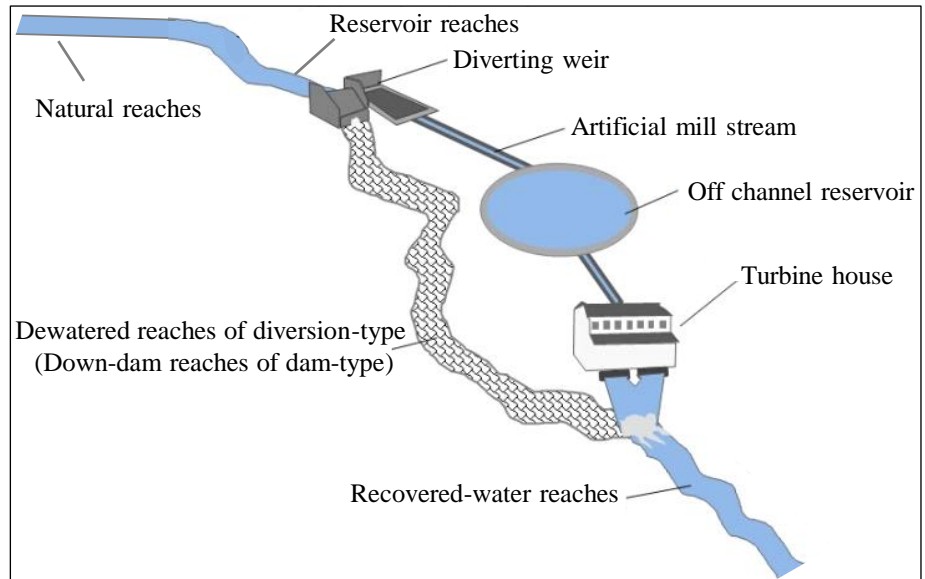

**Figure 2.** Schematic map of the diversion-type power station in the study (Modified from Guo et al. [3]).

The seven rivers investigated in this study were the (1) Yanshan River (YSR, Yanshan County, Jiangxi Province); (2) Jinggu River (JGR, Jinggu County, Yunnan Province); (3) Cheba River (CBR, Enshi City, Hubei Province); (4) Nan River (NR, Gucheng County, Hubei Province); (5) Chuanchang River (CCR, Nanjing County, Fujian Province); (6) Qingyang River (QYR, Shenlongjia County, Hubei Province); and (7) Bailong River (BLR, Zhouqu County, Gansu Province). Physical and hydrological data basic information and sampling details regarding the investigated rivers are shown in Table 1. A total of 133 samples were obtained from 7 rivers on which 45 hydropower stations are located.

**Table 1.** Physical and hydrological data and sampling details regarding the seven investigated rivers.

| River | Region | Discharge ($m^3$/s) | Length (km) | Catchment Area ($km^2$) | Mean Annual Temperature (°C) | Average Annual Rainfall (mm) | Investigated Station Number | Number of Sampling Sites/Sampling Time | Wet Season | Dry Season |
|---|---|---|---|---|---|---|---|---|---|---|
| Yanshan River (YSR) | Eastern | 60.2 | 87 | 1262 | 18.5 | 2094 | 8 | 11/2011-Nov | Apr.–Sep. | Oct.–Mar. |
| Chuanchang River (CCR) | Eastern | 32.6 | 121 | 1040 | 21.6 | 1876 | 5 | 11/2012-Mar | Apr.–Sep. | Oct.–Mar. |
| Jinggu River (JGR) | Southern | 15.5 | 85 | 634 | 20.2 | 1314 | 8 | 11/2011-Jul; 19/2012-May | May–Aug. | Sep.–Apr. |
| Cheba River (CBR) | Southern | 17.0 | 41 | 256 | 16.4 | 1425 | 7 | 9/2011-Sep | Jun.–Sep. | Oct.–May |
| Qingyang River (QYR) | Southern | 14.9 | 35 | – | 11.6 | 1170 | 5 | 12/2011-Nov; 13/2012-Jun | Jun.–Sep. | Oct.–May |
| Nan River (NR) | Southern | 79.3 | 255 | 6497 | 15.4 | 918 | 4 | 12/2011-Oct; 19/2012-Jun | May–Sep. | Oct.–Apr. |
| Bailong River (BLR) | Western | 389 | 570 | 31,800 | 12.7 | 434 | 8 | 16/2012-Nov | Jun.–Oct. | Nov.–May |

*2.2. Methods*

2.2.1. Sampling Program

In this study, macroinvertebrates in the rivers were collected using a nylon yarn D-frame net (width: 0.30 m, mesh size: 450 μm, sampling area: 0.15–0.30 m$^2$ of different sites), and the samples in the reservoirs were collected using a Peterson grab (1/16 m$^2$). Samples were sieved in situ, and 420 μm fractions were live-picked in the field and preserved in 10% formalin. Sorted samples (tissue dry mass for Mollusca) were weighed (wet weight, nearest 0.0001 g) using an electronic balance (Sartorius, Model BS224 S, Hamburg, Germany) to calculate the biomass. Macroinvertebrates were identified to the lowest possible taxon in the laboratory with the aid of a dissecting microscope by using identification keys [24–26]. All identified taxa were assigned to functional feeding groups (shredders, collectors, scrapers, and predators) following the definitions of Morse et al. [26] and Liang and Wang [27]. At each same sample sites, the river sediment had been collected, classified, and determined with a laser particle size analyzer (Mastersizer 3000, Malvine, UK). Moreover, water depth and velocity were obtained by means of field monitoring.

2.2.2. Data Analysis

The macroinvertebrate density (ind/m$^2$) and biomass (g/m$^2$) at each site were calculated using the arithmetic mean from all samples sites. PAST version 2 (Oslo, Norway) was employed to perform a nonparametric multivariate analysis of variance (PERMANOVA), and Canoco version 4.5 (Microcomputer Power, Ithaca, NY, USA) was used to conduct a gradient analysis (GA) to analyze the distribution characteristics of the community sites. The gradients used in the GA were derived from a detrended correspondence analysis (DCA). Excel 2013 (Microsoft Office 2013, Seattle, WA, USA) was used to perform the data analysis and plot charts for density, biomass, diversity index calculation, and taxa composition of macroinvertebrates, and all other statistical analyses were performed by SPSS version 17 (Chicago, IL, USA). One-way analysis of variance (ANOVA) was employed to analyze the significance of different habitat characteristics ($\alpha$ = 0.05).

2.2.3. Diversity Index Calculation Method

(1) The formula for calculating the Shannon–Wiener index [28] is as follows:

$$H' = -\sum_{i=1}^{S} P_i \log_2 P_i, \tag{1}$$

$$H'_{max} = \log_2 S, \tag{2}$$

where $P_i = n_i/N$, $n_i$ is the number of taxa $i$, $N$ is the total number of specimens, and $S$ refers to the number of taxa in the community.

(2) The formula for calculating the Margalef richness index [29] is as follows:

$$d = (S-1)/\ln N, \tag{3}$$

where $S$ denotes the number of taxa in the community and $N$ is the total number of specimens.

**3. Results and Discussion**

*3.1. Taxa Composition and Standing Crop*

3.1.1. Taxa Composition

A total of 133 macroinvertebrate samples were collected from the seven rivers, comprising 126 taxa belonging to 36 families and 72 genera. Taxa list of insects found in these rivers was shown in Table

S1 of supplementary materials. Insects represented the most diverse group, comprising 86 taxa. *Oligochaeta*, *Molluscs*, and other groups (*Nematoda*, *Hirudinea*, and *Crustacea*) comprised 22, 12, and 6 taxa, respectively (Figure 3A). *Diptera* was the dominant group among the aquatic insects (Figure 3B).

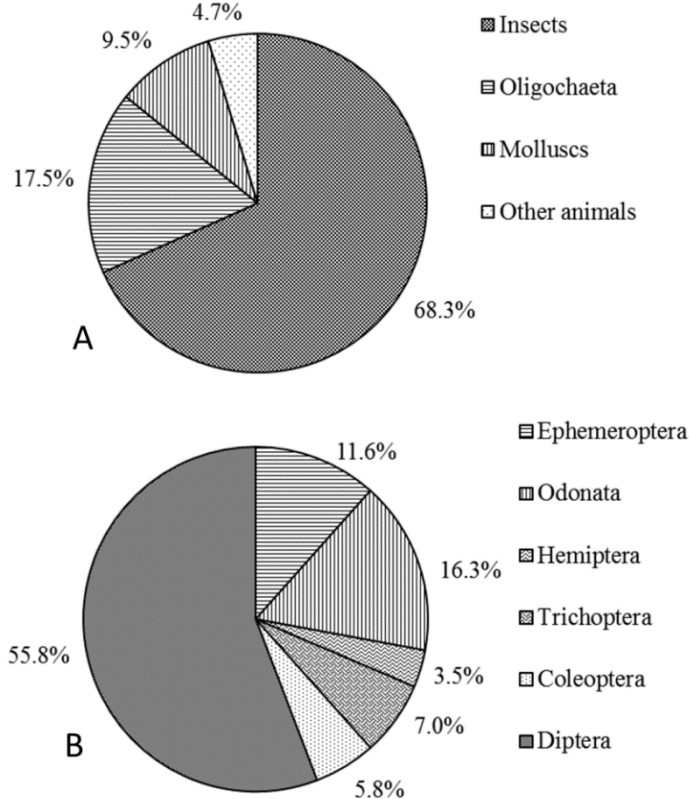

**Figure 3.** Taxa composition of (**A**) macroinvertebrates and (**B**) aquatic insects based on 133 samples.

The highest number of macroinvertebrate taxa was observed in YSR, which had an average of 16 taxa at each site, and the fewest taxa were present in BLR, with only four taxa observed at each site. In general, the number of macroinvertebrate taxa was higher in the more southern regions.

### 3.1.2. Standing Crop

Figure 4 shows that the average macroinvertebrate density and biomass in the seven rivers were $966 \pm 112$ ind/m$^2$ and $17.31 \pm 1.54$ g/m$^2$, respectively. Those in BLR were the lowest among the seven rivers ($566 \pm 71.12$ ind/m$^2$ and $2.22 \pm 0.35$g/m$^2$, respectively), whereas those in CCR were the highest among the seven rivers ($2079 \pm 247.68$ ind/m$^2$ and $53.98 \pm 7.72$ g/m$^2$, respectively). *Oligochaeta* was the dominant subclass in CCR, where the individual site density was higher than 8000 ind/m$^2$. In general, the macroinvertebrate density and biomass were higher in the more southern regions.

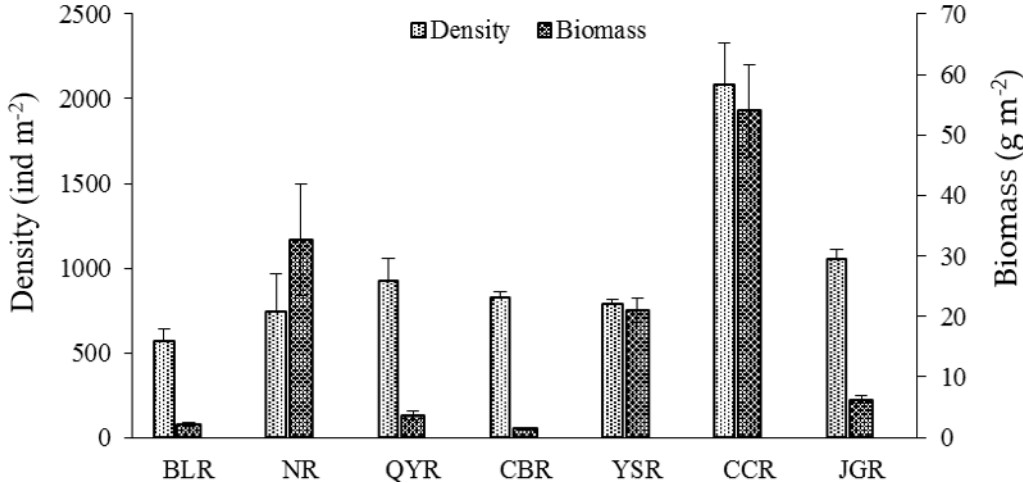

**Figure 4.** Average density and biomass of macroinvertebrates in the seven investigated rivers.

### 3.2. Effect of Small Hydropower Stations on Macroinvertebrates

All sample sites were classified into five habitat types (reservoir reaches, down-dam reaches, dewatered reaches, recovered-water reaches, and natural reaches) according to the sampling locations. All 126 taxa collected from these five habitat types were organized into an ordination diagram (Figure 5) through a DCA sequencing analysis. The results showed that the explanation rate for the taxa difference of Axis 1 was approximately 11.2%, the cumulative interpretation rate for Axes 1 and 2 was 18.3%, and the maximum gradient length was approximately 5.837, indicating that the taxa response to the environment exhibited a single peak pattern. The five habitat types are scattered throughout the ordination diagram, and they do not form a partition trend. This implies that the main factors affecting the distribution of macroinvertebrates in the different rivers and hydrological periods are not a result of runoff regulation; rather, the difference among the habitat types is due to temporospatial differences.

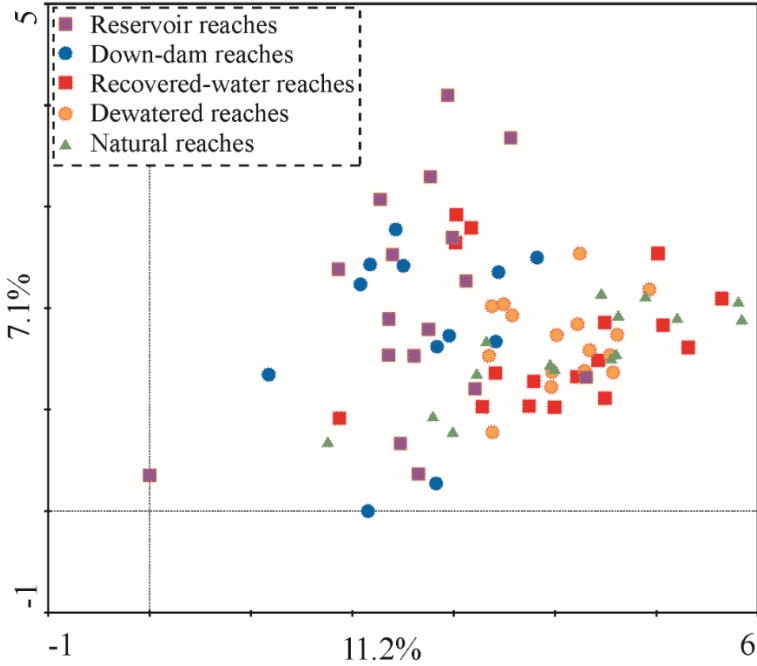

**Figure 5.** Ordination diagram of all sample sites through a detrended correspondence analysis (DCA) sequencing analysis.

### 3.2.1. Influence of Dam-Type Hydropower Stations on Macroinvertebrates

Four dam-type hydropower stations are situated along NR, making this river an effective example for illustrating the influence of dam-type hydropower on macroinvertebrates. The position of the sampling sites with respect to the hydroelectric station of NR was shown in Figure S1 from supplementary materials. Sediment grading curves from different sampling sites of NR was shown in Figure S3 from supplementary materials. Runoff regulation and the hydrological period (wet versus dry season) were found to be the two main flow regulation modes influencing macroinvertebrates in this river. Runoff regulation resulted in the formation of three habitat types (i.e., reservoir reaches, down-dam reaches, and natural reaches). Differences in the taxa abundance between the different habitat types were analyzed using single-factor ANOVA under two flow regulation modes (Table 2).

**Table 2.** One-way analysis of variance (ANOVA) results for macroinvertebrate abundance under two flow regulation modes (Runoff regulation and hydrological period) for Nan River (NR).

| Flow Regulation Modes | Statistical Groups | Sum of Squares | Df | Mean Square | F | Significance |
|---|---|---|---|---|---|---|
| Runoff regulation | Between groups | 81.04 | 2 | 40.52 | 9.46 | 0.001 ** |
| | In groups | 89.92 | 21 | 4.28 | – | – |
| | Total | 170.96 | 23 | – | – | – |
| Hydrological period | Between groups | 12.04 | 1 | 12.04 | 1.667 | 0.21 |
| | In groups | 158.92 | 22 | 7.22 | – | – |
| | Total | 170.96 | 23 | – | – | – |

Note: ** represents significant level at 1% with one-way ANOVA test. Df represents degree of freedom). F represents F value of one-way ANOVA test.

The results showed that for the taxa parameters, the runoff regulation mode had a significant effect whereas the influence of the hydrologic period was relatively weak; therefore, the hydrological data for 2011 (dry season) and 2012 (wet season) could be analyzed without considering the influence of different periods. PERMANOVA of the NR basin data showed that runoff regulation differed significantly in its influence on the different macroinvertebrate community structures ($p = 0.0036$, 99,999 times based on permutation test results). Figure 6 shows different sections of the macroinvertebrate DCA sequencing results.

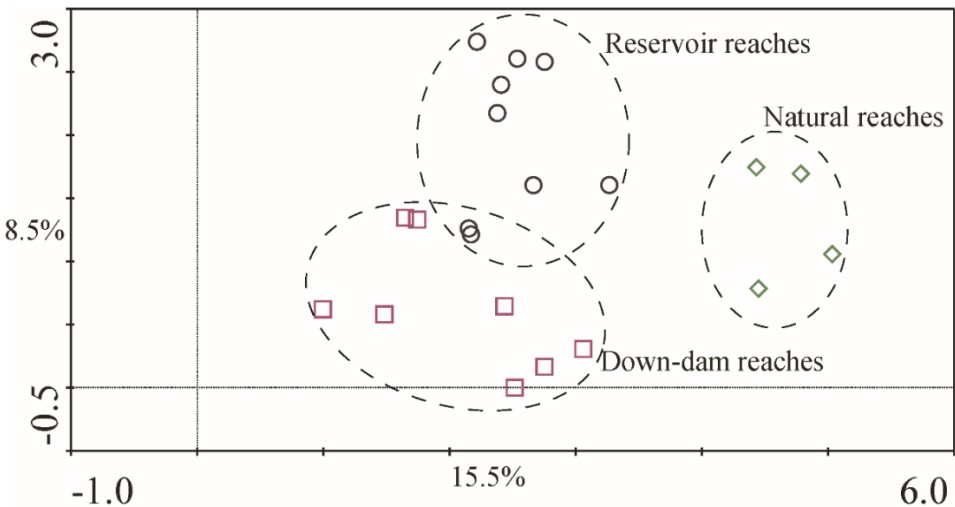

**Figure 6.** Ordination diagram of the macroinvertebrate from different sections in NR.

The explanation rate for the taxa difference of Axis 1 was approximately 15.5%, the cumulative interpretation rate for Axes 1 and 2 was 24.0%, and the maximum gradient length was approximately 5.033, indicating that the taxa response to the environment exhibited a single peak pattern. The three sample types formed obvious partition trends in the ordination diagram, and the grouping was basically consistent with the runoff regulation model. This indicated that, regarding the distribution characteristics of taxa communities among the environmental gradient in the NR area, significant differences were observed among the sites at the dam-type hydropower station, down-dam reaches, and natural reaches. The difference in organisms between the reservoir reaches and down-dam reaches was mainly along Axis 2, and the difference between the natural reaches and unnatural reaches was mainly along Axis 1.

The structure parameters of the macroinvertebrate communities in the reservoir area, dam, and natural river reaches were classified and counted (Table 3). The results showed small deviations and fluctuations in taxa abundance and diversity (as indicated by the Shannon–Wiener and Margalef richness indices), indicating that the three parameters for the groups at each sampling section were relatively consistent. By contrast, the volatility in deviations of density and biomass was more pronounced, indicating significant differences in these parameters between these groups. Taxa richness gradually increased from reservoirs to down-dam reaches and then to natural reaches (4.4, 6.5, and 9.5, respectively). The Shannon–Wiener index also exhibited an increasing trend in these areas (1.06, 1.48, and 1.58, respectively), whereas the biomass exhibited a decreasing trend (56.3, 25.2, and 6.0 g/m$^2$, respectively). According to these parameters, the characteristics of the community structure in three typical areas can be summarized as follows. In reservoirs, the number and diversity of macroinvertebrate taxa were the lowest, with higher concentrations of individual taxa, although individual animals were heavier. In down-dam reaches, the number and diversity of macroinvertebrate taxa increased slightly, but the biomass was not high because of the low weight of individual animals. In natural reaches, the number and diversity of macroinvertebrate taxa were the highest, and the weight of individual animals was the lowest.

**Table 3.** Macroinvertebrate structure parameters (Mean ± SD, SD stands for standard deviation) in reservoirs, down-dam reaches, and natural reaches in NR.

| Community Parameters | Reservoir | Down-Dam Reaches | Natural Reaches |
|---|---|---|---|
| Taxa abundance | 4.4 ± 0.7 | 6.5 ± 0.3 | 9.5 ± 1.3 |
| Density (ind/m$^2$) | 412.0 ± 132.3 | 1396.0 ± 733.6 | 825.4 ± 160.3 |
| Biomass (g/m$^2$) | 56.3 ± 20.9 | 25.2 ± 11.4 | 6.0 ± 2.6 |
| Shannon–Wiener index | 1.06 ± 0.16 | 1.48 ± 0.07 | 1.58 ± 0.21 |
| Margalef richness index | 6.22 ± 1.05 | 9.66 ± 0.63 | 10.46 ± 1.16 |

According to their mode of ingestion, macroinvertebrates were classified into four functional feeding groups: scrapers, predators, filter-collectors, and shredders. The statistical results for the functional feeding groups in the reservoir, down-dam reaches, and natural river reaches indicate that the proportion of collectors in each group accounted for only a small deviation, showing that the number of individual taxa was relatively stable (Table 4). From the reservoir to the down-dam reaches and onto the natural river reaches, the percentage of collectors exhibited a decreasing trend, indicating a single type of organism in the reservoir (collectors), which mainly fed on organic fine particles and plankton. Because of environmental changes to the habitat, shredders and scrapers accounted for a larger part in the down-dam reaches. In terms of quantity, the increase in the proportion of shredders correlated with the increase in organic fine particles. This increase in the proportion of scrapers indicated that the down-dam environment was suitable for attached algae, which contributed to its growth and development. The proportion of scrapers and predators in the natural river reaches was high, whereas that of various types of feeding organisms was relatively balanced. These results indicated that more biological types were present in the natural river reaches and that functional feeding patterns tended to be diversified.

**Table 4.** Percentage (Mean ± SD) of different functional feeding groups of macroinvertebrates in the NR reservoir, down-dam reaches, and natural reaches.

| Functional Feeding Groups | Reservoirs | Down-Dam Reaches | Natural Reaches |
|---|---|---|---|
| Collector | 76.8% ± 5.2% | 55.6% ± 4.8% | 44.4% ± 8.9% |
| Shredder | 6.6% ± 2.4% | 22.7% ± 4.9% | 3.8% ± 3.0% |
| Scraper | 10.8% ± 4.1% | 20.0% ± 4.3% | 36.4% ± 14.4% |
| Predator | 5.9% ± 2.3% | 1.6% ± 0.9% | 15.3% ± 7.1% |

The macroinvertebrates were comprised of *Arthropoda*, *Annelida*, and *Mollusca* in the NR. The proportion of individual *Arthropoda* taxa was the highest (57.6% ± 6.8%), followed by *Mollusca* taxa (30.4% ± 6.5%), and then *Annelida* taxa (12.0% ± 4.4%). *Arthropoda* proportions of individual macroinvertebrate taxa in the reservoirs, down-dam reaches, and natural reaches were 37.2% ± 9.0%, 69.3% ± 7.3%, and 95.5% ± 2.1%, respectively, *Mollusca* were 9.0% ± 43.4%, 7.3% ± 24.7%, and 2.1% ± 2.6%, respectively, and *Annelida* were 43.4% ± 10.5%, 24.7% ± 7.7%, and 2.6% ± 2.6%, as shown in Table 5. In reservoirs, *Arthropoda* taxa accounted for the lowest proportion among the three habitat types, whereas the proportion of *Mollusca* taxa was the highest (although the overall proportion was still low). In the down-dam reaches, the proportion of *Arthropoda* taxa was higher relative to that in the reservoirs, whereas the proportion of *Mollusca* and *Annelida* taxa was lower. In the natural reaches, the proportion of *Arthropoda* taxa was the highest among the three habitat types, and those of the *Annelida* and *Mollusca* taxa were the lowest among the habitat types.

**Table 5.** Proportion (Mean ± SD) of different individual macroinvertebrate taxa in the reservoirs, down-dam reaches, and natural reaches, stratified by phylum in NR.

| Classification | Reservoir | Down-Dam Reaches | Natural Reaches |
|---|---|---|---|
| *Arthropoda* | 37.2% ± 9.0% | 69.3% ± 7.3% | 95.5% ± 2.1% |
| *Mollusca* | 9.0% ± 43.4% | 7.3% ± 24.7% | 2.1% ± 2.6% |
| *Annelida* | 43.4% ± 10.5% | 24.7% ± 7.7% | 2.6% ± 2.6% |

Among the *Arthropoda* taxa, aquatic insects accounted for a high proportion of individual taxa, with the arthropod organisms in most samples comprising entirely aquatic insects. In the reservoirs and down-dam reaches, the midges (*Diptera*, *Chironomidae*) were the absolute dominant taxa among the aquatic insects, accounting for nearly 100% of *Diptera* taxa. In the natural reaches, the presence of aquatic insects from orders other than *Diptera*, including *Ephemeroptera* (76% ± 5.8%), *Odonata* (1.5% ± 1.1%), *Trichoptera* (6.6% ± 2.1%), and *Coleoptera* (1.5% ± 0.9%), taken together with those from *Diptera* (14.3% ± 4.9%), also confirmed that diverse organism types were present in the natural reaches. Concurrently, the process of artificial reservoir flow regulation altered the environmental conditions (e.g., flow regime and sediment composition) and habitat factors, causing considerable changes in the taxa composition and community structure of aquatic insects.

3.2.2. Influence of Diversion-Type Hydropower Stations on Macroinvertebrates

To analyze the influence of diversion hydropower stations on macroinvertebrates community structure, data on the dewatered reaches, recovered-water reaches and natural reaches for 2011 (wet season) and 2012 (dry season) in QYR, where there are many diversion-type hydropower stations, were selected for analysis. The position of the sampling sites with respect to the hydroelectric station of QYR was shown in Figure S2 from supplementary materials. Sediment grading curves from different sampling sites of QYR was shown in Figure S4 from supplementary materials. One-way ANOVA was adopted to test for significant differences in taxa abundance (at the 0.05 level) under the two types of classification, which were runoff regulation and hydrological period. The results showed that the effect of the hydrological period was more significant than that of runoff regulation, which was relatively weak (Table 6). Table 7 presents the macroinvertebrate structure parameters for QYR for the wet and

dry seasons. The table data show an increase in macroinvertebrate taxa abundance (5.2 to 8.3), density (322.2 to 1170.5 ind/m²), and biomass (24.6 to 40.1 g/m²), as well as increases in Shannon–Wiener index (1.23 to 2.08) and Margalef richness index (1.66 to 3.51).

**Table 6.** One-way ANOVA results for macroinvertebrate abundance under two flow regulation modes in Qingyang River (QYR).

| Flow Regulation Modes | Statistical Groups | Sum of Squares | Df | Mean Square | F | Significance |
|---|---|---|---|---|---|---|
| Runoff regulation | Between groups | 6.81 | 2 | 3.40 | 0.24 | 0.793 |
| | In groups | 318.63 | 22 | 14.83 | – | – |
| | Total | 325.44 | 24 | – | – | – |
| Hydrological period | Between groups | 217.50 | 1 | 217.50 | 46.34 | 0.000 ** |
| | In groups | 107.94 | 23 | 4.69 | – | – |
| | Total | 325.44 | 24 | – | – | – |

**Table 7.** Macroinvertebrate community structure parameters (Mean ± SD) in the wet and dry seasons in QYR.

| Community Parameters | Wet Season | Dry Season |
|---|---|---|
| Taxa abundance | 5.2 ± 0.5 | 8.3 ± 0.6 |
| Density (ind/m²) | 322.2 ± 92.8 | 1170.5 ± 131.7 |
| Biomass (g/m²) | 24.6 ± 5.1 | 40.1 ± 10.8 |
| Shannon–Wiener index | 1.23 ± 0.15 | 2.08 ± 0.12 |
| Margalef richness index | 1.66 ± 0.45 | 3.51 ± 0.87 |

PERMANOVA of the macroinvertebrates in QYR also showed that based on the runoff regulation grouping, the difference between macroinvertebrate community structures was nonsignificant in 2011 ($p = 0.169$) and 2012 ($p = 0.191$). The DCA results for 2011 (Figure 7) and 2012 (Figure 8) in QYR indicated a high degree of similarity in the samples obtained from dewatered reaches, recovered-water reaches, and natural reaches (Figures 7 and 8). These regions overlap each other and showed no obvious distinction. This was consistent with the ANOVA and PERMANOVA results.

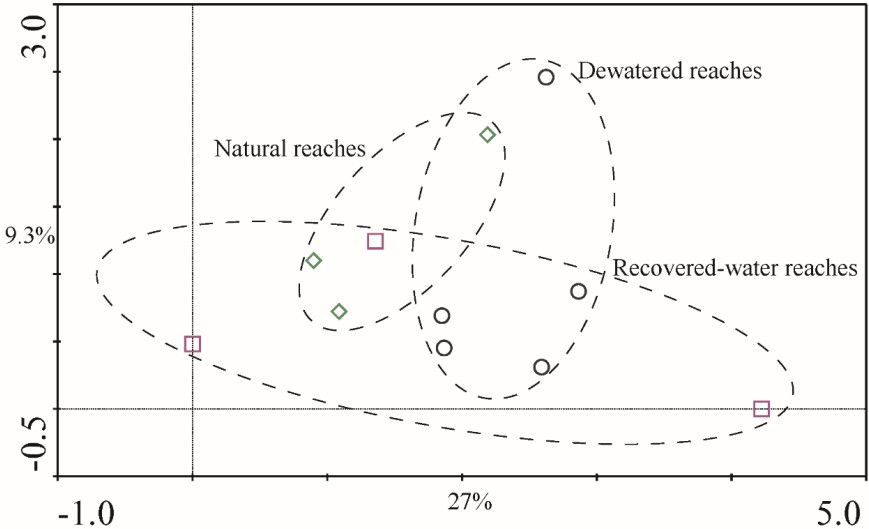

**Figure 7.** Ordination diagrams of macroinvertebrates under the runoff regulation grouping from QYR in 2011.

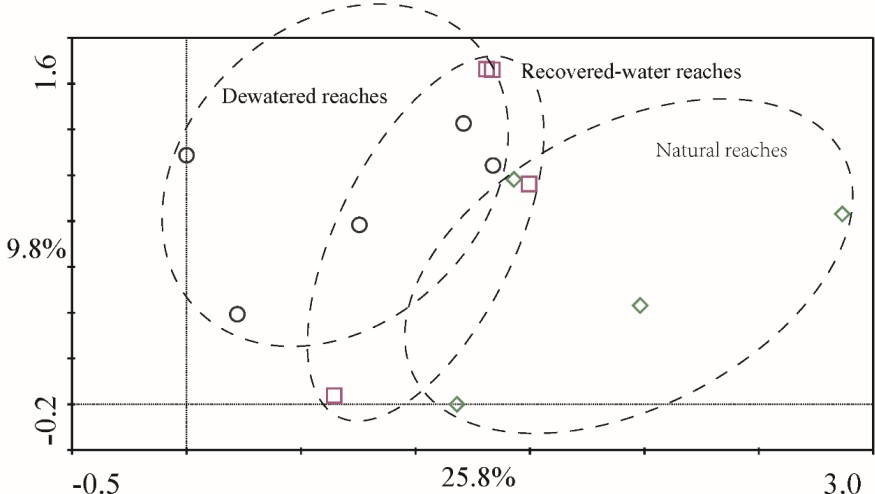

**Figure 8.** Ordination diagrams of macroinvertebrates under the runoff regulation grouping from QYR in 2012.

In summary, the influence of hydropower stations on macroinvertebrate communities exhibited obvious differences between the dam- and diversion-type power stations. For dam-type hydropower stations, a reservoir forms because of the high dam. The substrate, velocity, and depth in reservoirs (silt substrate, low velocity, and high depth) differ from those in down-dam reaches (gravel substrate, high velocity, and small depth) and natural reaches. Hence, the macroinvertebrate taxa composition, standing crops, diversity, and functional feeding groups exhibited clear differences between reservoirs and down-dam reaches. These results are similar to those reported by Ren et al. [30]. Furthermore, the hydrological period had no influence on these factors, as evidenced by the habitat characteristics of the reservoirs and down-dam reaches exhibiting no change with the hydrological period.

For diversion-type power stations, no obvious reservoir formed because of the low dam; however, dewatered reaches and recovered-water reaches did form as a result of these power stations being installed. However, there was no significant substrate difference among dewatered reaches, recovered-water reaches, and natural reaches with the same type of gravel. Many studies have shown that sediment is one of the most crucial factors influencing macroinvertebrates [31–34]. Identical substrate types also indicate similar macroinvertebrate structures. Nevertheless, the intercepting effect of diversion-type power stations (low dam) in the wet season was not apparent. The dewatered reaches and recovered-water reaches would have been eroded as a result of flooding, which could have been linked to the low level of macroinvertebrate standing crops. Therefore, the macroinvertebrates near diversion-type power stations were susceptible to hydrological period effects [35]. One extreme impact on macroinvertebrates near diversion-type power stations is zero flow and dry ups. This phenomenon occurs occasionally in the mountainous rivers in the dry season with a lack of a runoff supply. In this situation, most macroinvertebrate species may disappear and community structure will change in the river.

## 4. Conclusions

This study was conducted to investigate the influence of small hydropower stations on macroinvertebrate communities in mountainous rivers in the Changjiang River Basin of China. In summary, the influence of hydropower stations on macroinvertebrate communities exhibited obvious differences between the dam- and diversion-type power stations. For dam-type hydropower stations, the intercepting effect of the dam was the main influencing factor, whereas the influence of the hydrological period was nonsignificant. Macroinvertebrates in reservoirs (up-dam), down-dam reaches, and natural reaches exhibited an obvious zoning phenomenon due to differences in the substrate and hydraulic conditions between reservoirs, down-dam reaches, and natural reaches.

For diversion-type hydropower stations, the hydrological period was the main influencing factor, whereas the intercepting effect of the dam was nonsignificant. Macroinvertebrate abundance in dewatered reaches, recovered-water reaches, and natural reaches exhibited no obvious gradient difference in the same hydrological periods because the substrate composition was identical. However, the macroinvertebrate density, biomass, and diversity clearly differed between the wet and dry season due to the different hydraulic conditions. Thus, substrate composition and flow regulation were the main factors influencing the macroinvertebrate structure in these rivers.

**Supplementary Materials:** The following are available online at http://www.mdpi.com/2073-4441/11/9/1892/s1, Table S1. Taxa list of insects that are found in these rivers. Figure S1. The position of the sampling sites with respect to the hydroelectric station of NR. Figure S2. The position of the sampling sites with respect to the hydroelectric station of QYR. Figure S3. Sediment grading curves from different sampling sites of NR. Figure S4. Sediment grading curves from different sampling sites of QYR (NNM-S represents Niangniangmiao station).

**Author Contributions:** Data curation, W.G.; formal analysis, W.Z. and L.Z.; investigation, W.Z., W.G., L.Z., and X.C.; project administration, Q.L.; resources, Q.L.; supervision, X.T.; writing—original draft, W.Z.

**Funding:** This research was financially supported by the National Key Research and Development Program (Grant No. 2017YFC0404502 and 2018YFC0407603), and the fund for Basic Scientific Research Business of Central Public Research Institutes (Grant No. CKSF2019292/SH, CKSF2017026/SH, CKSF2019513/SH and CKSF2019251/SH).

**Acknowledgments:** We thank the reviewers for their useful comments and suggestions.

**Conflicts of Interest:** The authors declare no conflict of interest.

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
