# Peer review of "Influence of Different Types of Small Hydropower Stations on Macroinvertebrate Communities in the Changjiang River Basin, China"

_water, doi:10.3390/w11091892_

Round 1

Reviewer 1 Report

The authors studied the influence of different types of hydropower plants on MZB. The results are weel exposed and the work is interesting given that one of the most importat problem in ecohydraulics is to understand if the MZD can be used as target to quantify the water quality near a hydropower plant. The english is good. I have only some suggestions. 1. Improve image quality 2. in the introduction at the end of the sentence "171-km reach in the United States." You can add the following sentence and reference "Premstaller et al. used the fish and macroinvertebrates to quantify the effects of a big reservoirs in Italy and the effects of hydropeaking downstream the dam". Georg Premstaller, Valentina Cavedon, Giuseppe Roberto Pisaturo, Steffen Schweizer, Vito Adami, Maurizio Righetti, Hydropeaking mitigation project on a multi-purpose hydro-scheme on Valsura River in South Tyrol/Italy, Science of The Total Environment, Volume 574, 2017, Pages 642-653, ISSN 0048-9697, https://doi.org/10.1016/j.scitotenv.2016.09.088. 3. In table 1 add the acronym for each river 4. Shannon-Wiener index (add reference) 5. Margalef (add reference)

Reviewer 2 Report

Summary and Broad Comments for Authors

Although numerous studies have examined the impact of hydropower stations on aquatic macroinvertebrates, few have compared the effects of different types of hydropower stations on aquatic macroinvertebrates. The aim of this study was to fill that void by comparing the effects of dam-type hydropower stations and diversion-type hydropower stations in southern China. The authors collected macroinvertebrate samples from seven rivers over a three-year period. In focusing on two of the rivers, the authors reported differences between the two types of hydropower stations. In dam-type stations the dam with its resulting reservoir was the primary influence on macroinvertebrate populations while hydrological period was not significant. Species richness and Shannon-Wiener index scores increased along a downstream continuum from reservoir to down-dam reaches to natural reaches while biomass decreased along the same continuum. In diversion-type dams, hydrological period was the primary influence on macroinvertebrate populations while the effect of the dam was not significant. Macroinvertebrate abundance, density, biomass, and Shannon-Wiener index scores increased from wet to dry seasons. The manuscript is reasonably well written and reveals the authors’ decent command of the English language. However, the study primarily reports on data from within two of the rivers (NR and QYR) and, consequently does not adequately compare the trends among all of the sampled rivers. The manuscript would also benefit from a more thorough review of the literature and clarification of ambiguous statements or expansion of content to provide necessary details on the methodology employed. As per the journal’s recommended guidelines for reviewers, please see the broad comments provided below as well as the specific line-item comments that follow.

Originality/Novelty: Is the question original and well defined? Do the results provide an advance in current knowledge? Given the dearth of studies analyzing the effects of different types of hydropower stations on aquatic macroinvertebrate populations, the idea is original and can add valuable information to the existing literature. The manuscript will benefit from a more thorough review of the literature, specifically in the Introduction to establish the claim that previous studies have demonstrated that changes in river morphology and hydraulic conditions influence benthic community structure. Similarly, the originality of the study’s focus can be improved by sharing the findings of the few studies that have previously explored the effects of different types of hydropower stations on macroinvertebrate populations. The authors cite several studies that examine the effects of hydropower stations in general on macroinvertebrates and contrast that by stating that “few studies” have explored the effects of different types of hydropower stations. However, the authors neglect to cite or share findings from these “few studies.” Doing so would help to establish the rarity of such studies, thereby highlighting the originality of this study. Finally, the research question is only implied, rather than being well defined and explicitly stated, and no hypotheses are given. This makes it difficult to fairly evaluate the findings reported later in the manuscript.

Significance: Are the results interpreted appropriately? Are they significant? Are all conclusions justified and supported by the results? Are hypotheses and speculations carefully identified as such? Although the authors state that they collected data from seven rivers, analysis concentrates primarily on trends within two of the rivers (NR and QYR) leaving very little comparison among the seven rivers. The only justification offered for this selective use of data is that NR includes four dams and QYR has many diversions. The authors did not clearly specify their hypotheses and did not provide important details in the Methods section to inform the reader about each sampled river. For instance, which types of hydropower stations were found along each sampled river, how many of these hydropower stations were found along each sampled river, how many samples were collected from each river, and where in relation to the hydropower stations were samples collected in each river? Also, the response variables the authors aim to analyze are only hinted at in describing the equipment or software used to collect data, rather than explicitly stated. Given the lack of clearly stated research questions or hypotheses in the Introduction, the lack of important information in the Methods, and the very limited comparison of data among the seven rivers, the reported results are too incomplete to fairly determine what trends might exist or to draw broad conclusions. The goal of the study as implied in the Introduction was to determine the effects of two different types of hydropower stations on macroinvertebrate populations. Consequently, it would seem to make more sense to identify the rivers or reaches with dam-type hydropower stations and the rivers or reaches with diversion-type hydropower stations and compare the data among these two categories.

Quality of Presentation: Is the article written in an appropriate way? Are the data and analyses presented appropriately? Are the highest standards for presentation of the results used? The authors demonstrate a decent command of the English language, so I have little reservation regarding the readability of what is presently written. However, in several places throughout the manuscript (see suggestions above and below as well as the specific line-item comments) statements are ambiguous or lack sufficient support, which compromises the manuscript’s overall quality of presentation. In particular, the Methods section lacks important details regarding the study design, sampling protocol, and statistical analysis. Also, the results and discussion are presently combined in a single section. Although this may be acceptable in some publications, the conventional practice is to present a separate Results section followed by a separate Discussion section. I believe dividing the findings into separate Results and Discussion sections is preferable because readers are more familiar with this practice, and it separates the information into easier-to-read sections. I encourage the authors to restructure the manuscript to account for these recommendations.

Scientific Soundness: is the study correctly designed and technically sound? Are the analyses performed with the highest technical standards? Are the data robust enough to draw the conclusions? Are the methods, tools, software, and reagents described with sufficient details to allow another researcher to reproduce the results? It is difficult to fairly evaluate the soundness of the study due to the lack of clearly stated research questions or hypotheses and the lack of details provided in the Methods section. Specifically, important details regarding the sampling sites and sampling protocol are lacking (see general comments above and specific line-item comments) and no overall study design is identified (i.e., Does it follow a BACI approach, etc?). In describing their results, the authors mention environmental conditions such as flow regime and sediment composition, but do not explain how or when these were observed or measured, nor do they provide a general description of physical and hydrological features of their study sites in the Methods section. This lack of information also compromises the ability of other researchers to reproduce the methodology. Similarly, some valuable details regarding statistical analyses are lacking. For example, the authors only generally state that Excel and SPSS were used for statistical analysis, but are not clear on how Excel was used. The use of Excel for data summaries and creation of plots is acceptable, but relying on Excel’s built-in data analysis tools for statistical tests is questionable for a scientific study given the errors that have been documented in the literature (see McCullough and Wilson, 2002) and the availability of other more robust tools. The authors need to specifically state how and what Excel was used for vs. what SPSS was used for.

Interest to the Readers: Are the conclusions interesting for the readership of the Journal? Will the paper attract a wide readership, or be of interest only to a limited number of people? (please see the Aims and Scope of the journal) The study is of interest because of the ubiquity of hydropower stations and the usefulness of aquatic macroinvertebrates as indicators of environmental impact. Providing clarification and expansion of content as recommended above will improve readers’ ability to understand the significance of the study and, hence, increase likelihood of expanded readership. Providing more detailed information on the study design and sampling protocol and redoing the analyses to more thoroughly compare data from among all seven sampled rivers would reveal overall trends in the sampling region and enable valid conclusions to be drawn, which can broaden interest among readers.

Overall Merit: Is there an overall benefit to publishing this work? Does the work provide an advance towards the current knowledge? Do the authors have addressed an important long-standing question with smart experiments? Overall, the study has potential to contribute valuable findings regarding the impact of different types of hydropower stations on aquatic macroinvertebrates. In conducting this study, the authors have addressed a topic that has not received much prior attention and can therefore advance the body of scientific knowledge. However, as currently reported the study does not reveal adequate study design, thorough methodology, or sufficient analysis. Providing the necessary information in the Methods section and redoing the analysis to account for all of the sampled rivers to adequately determine differential effects of dam-type stations and diversion-type stations on macroinvertebrate populations could help amend this flaw.

English Level: Is the English language appropriate and understandable? The authors possess decent command of the English language as demonstrated by good choice of wording and sentence structure that enables the manuscript to flow logically from introduction to conclusion. The reader’s ability to understand the work and its applicability is presently limited only by the incomplete information and analysis as detailed above and in the line-item comments.

Specific Comments for Authors

Line 20 – “station” should be plural

Line 21 – You mention 126 species, but if all were not identified to the species level it would be more appropriate to refer to them as taxa. Also “macroinvertebrate” should be plural here.

Line 37 – I recommend updating this statistic to account for the status of hydropower stations as of 2019. Also, include a citation so the reader knows where this statistic is from.

Line 39-43 – Good job of defining the hydropower station types.

Line 45 – You mention “previous studies” but only cite one. Surely there are other studies that also demonstrate the changes you mention. I encourage you to locate them and cite them here to validate the claims.

Line 48 – Define “small-scale water ecosystems” by providing criteria to differentiate small-scale from other scales as you did for the different types of hydropower stations above.

Line 50 – I question whether it is correct to refer to macroinvertebrates collectively as a “taxa.” Macronivertebrates is more of a catch-all term rather than a taxonomic term. Also, your reason for highlighting this group might be improved by differentiating it from other taxa that are found in river habitats.

Line 56 – “ecological environment effects” seems redundant. It would suffice to refer to this as “ecological effects” or “environmental effects.”

Line 59 – “the macroinvertebrates indicator” should be “the macroinvertebrate indicators”

Line 67 – The meaning of the sentence here is unclear. A causal relationship is something that a study is designed to reveal, rather than something that would be apparent ahead of time. Rephrase to clearly explain the point you are intending to make.

Line 71 – You mention the “few studies” that have focused on the effects of different types of hydropower stations, but neglect to cite these studies and/or to explain the findings from them. These studies will be important to highlight and summarize in the Introduction to enable comparisons to your own study.

Line 76 – The choice of wording here is awkward. A study doesn’t select rivers; rather, researchers select rivers for inclusion in a study. Rephrase to improve clarity.

Line 84-85 – What work was previously done in these rivers? If it has been published, it would be helpful to cite it here so the reader understands the relevance to the present study. If it is not relevant to the present study, I question your justification for selecting these seven rivers.

Line 85 – “filed” should be “field”

Line 86 – insert “are” in front of “distributed”

Line 86 – You have not provided any description of the physical or hydrological features of the region or the seven rivers included in the study. This information needs to be included so the reader understands the similarity (or differences) among the selected rivers.

Line 88 – Which parts of the rivers were investigated? How long was each study reach? Where did each reach occur along the river continuum in relation to the hydropower stations? This is crucial information without which the study cannot be properly understood or evaluated.

Line 89 – Provide criteria (i.e., months of the year) to differentiate the wet and dry seasons as you did above in differentiating between the types of hydropower stations.

Line 97 – I recommend using a more specific term rather than “basic information.” Perhaps “physical and hydrological data” would be more suitable.

Line 99 – Same as the preceding comment. Also, provide the source for these data. Also, discharge is typically reported in cubic meters per second, so your header in the first column needs to be amended.

Line 102 – When were the samples collected? How many were collected at each site?

Line 107 – Offer a fuller explanation on what you mean by “lowest possible taxon.” Do you mean you identified organisms only to the level that your identification skills allowed or do you mean that you took each organism to the lowest taxonomic level that is known?

Line 108 – I question why the citations for 17 & 18 are inserted where they are. Why not include them with 19 at the end of the sentence, since they are all identification guides?

Line 109 – Here you identified four functional feeding groups, but in line 215 you have split the collectors into gatherers and filterers. Be consistent throughout the manuscript.

Lines 112-119 – It would be helpful to the reader here to explicitly state the response variables you intended to focus on, then go on to explain how you tested them.

Lines 116-117 – State clearly which tests Excel was used for and which tests SPSS was used for.

Line 122 – Cite a source for this formula. Perhaps Magurran, 2004 would be helpful.

Line 127 – Same as preceding comment.

Lines 133-134 – Since insects were the most abundant group, it would be helpful to the reader to include here in the text and/or in a table a list of the most abundant taxa. For example, list the five most abundant species (and include the family name) to give the reader a sense for the taxa that are found in these rivers. A primary focus of your study is that macroinvertebrates can be indicators of disturbance, but sensitivity to disturbance sometimes differs by the genus or species. Without such a list your reason for claiming that organisms were identified to the lowest possible taxon seems superfluous, since Figure 2 only goes to the order level.

Lines 146-147 – You make a reference to the substrate here, but you did not explain here or in the Methods section how you assessed substrate type or stability.

Lines 148-149 – Elaborate on why you think the larger macroinvertebrate density and biomass in CCR was due to high pollution. This seems counterintuitive unless these are pollution-tolerant taxa that have proliferated due to the absence of sensitive species. This affirms the importance of providing the reader with a list of the most common species in your study. Also, this segment would be more appropriate in the Discussion section. (See the broad comments for my recommendation to break the Results and Discussion into separate sections.)

Line 151 – Either here or in the Methods section, you need to inform the reader as to which regions you consider “southern” and which regions are in other categories (i.e., “northern,” etc.).

Line 152 – Have you conducted analyses to determine whether the differences in macroinvertebrate density and biomass among the seven rivers were significantly different or is this figure merely reflecting natural variation? If the former, insert labels to indicate where significance was found. If the latter, the figure offers little help toward understanding whether hydropower stations cause significant impact on macroinvertebrate populations.

Line 155 – Were these five categories determined before or after the sampling was conducted and what were the criteria used to categorize the sites? It would be helpful to the reader if you also provided a figure showing where each of these categories occurs along the river continuum within a study reach. Also, how many samples were taken from within each category? This is information that should be initially presented in the Methods section when you present your study design, sampling protocol, and plans for analysis.

Line 168 – Here you begin a long focus on NR but this diverges away from the intended aim of the study, which was to examine the differential impacts of the two different types of hydropower stations among the seven rivers you sampled. By focusing on a single river the data from the other rivers becomes obsolete and does not allow comparison. Also, if you report on data from within only a single river you need to account for pseudoreplication and explain in the Methods section how you have accounted for this in your statistical tests. Upstream influences can have downstream effects.

Line 196 – It would be helpful to the reader if you included a list of the structure parameters in parentheses here. This will prevent confusion over whether you are referring to stream segments or response variables. Also, stay consistent throughout the manuscript and use “down dam” rather than just dam to refer to these stream segments.

Line 215 – See previous comment in line 109.

Lines 223-228 – This is discussion so should be included in your Discussion section. Also, there is some ambiguity in the wording here. Specifically, 1) it is isn’t clear what the “its” refers to and 2) the phrase “various types of feeding organisms” is vague.

Line 231 – “macroinvertebrate” should be plural

Line 235 – “showed” should be “shown”

Lines 235-237 – This sentence is awkwardly phrased such that it is not in agreement with the data presented in Table 5. I encourage you to rephrase it to reflect what is actually shown in Table 5.

Line 242 – Does Table 5 refer to just NR or to all seven rivers? This should be made clear in the caption.

Lines 244-245 – You refer to species here but have not identified any species in the study. I recommend either using “taxa” here or elaborate on the insect species that were found, as recommended above in the comment for lines 133-134.

Line 246 – Chironomidae should be capitalized. Also, this is a type of midge, rather than a mosquito.

Line 252 – Were these parameters measured? (See similar comment in lines 146-147.

Line 255 – Here you begin a long focus on QYR. See similar comment in line 168. Also, “large-scale spatial factor” is somewhat vague. It would be helpful to more specifically define what you mean.

Line 257 – The wet and dry seasons have not been defined (i.e., what months comprise each season?). Please do so in the Methods section where you unveil your sampling protocol. If necessary, you can reiterate here.

Line 270 – The bottom line in Table 7 needs adjusting. Part of it is bolded and part is not bolded.

Lines 285-287 – You reference the streamflow and substrate characteristics as in earlier lines (see previous comment), but still need to provide some explanation for how these factors were assessed.

Lines 297-298 – Provide some citations to validate the relationship between macroinvertebrates and substrate.

Lines 298-299 – See previous comment in lines 285-287.

Lines 304-305 – When does this phenomenon occur?

Lines 305-306 – Your statement here is too broad, too definite, and without any citations to validate it. Some organisms are likely to die in such conditions, but others will leave and go to another aquatic system. Macroinvertebrates may migrate down into the hyporheic zone until favorable conditions return. The duration of the zero flow event matters too.

Line 325 – “Special’ appears to be unnecessarily capitalized here

Reviewer 3 Report

COMMENTS FOR THE AUTHORS

Manuscript by  WeiHua Zhao, XianQiang Tang,*, WeiJie Guo, Liangyuan Zhao, QingYun Li and XiaoHuan Cao

Entitled: “Influence of Different Types of Small Hydropower Stations on Macroinvertebrate Communities in Southern China” by Zhao et al.

GENERAL COMMENTS:

 The manuscript investigates the influence of different kind of small hydropower stations on macroinvertebrate communities in China.

The study is interesting because information on this topic is limited.  The ecological impact of small hydroelectric power plants could be mitigated with sustainable management thanks to these studies.

However, the experimental design is not very clear, it is not clear how many sites were monitored for each river, their position with respect to the hydropower stations, and when the samplings were carried out in the two periods. All these issues should be specified more clearly.

Also the macroinvertebrates sampling is not clear (see also minor comments).

The most important thing concerns the reservoirs that are compared to river reaches! The macrobentonic community of a reservoir is completely different with respect to a river reach, this must be clearly highlighted in the text.

The applied methodology is the classical one, however the simplicity of the indices, in my opinion, provides an interpretative clarity, much appreciated by the readers.

MINOR COMMENTS:

 - replace “species” with “taxa” in all the sections

- in the text it is necessary to indicate the number of sampling stations, from Tab. 1 they should be 48

Abstract

Lines 20-21: from section 2 the sampling period seem to be 2011-2013 while in Tab. 1 is 2011-2012 as at lines 83-84, clarify this aspect

Line 22: delete “arthropod”

Keywords

Replace the keywords “different types” and “influence”, they are not appropriate.

Keywords must give a clear idea of the topic!

Introduction

Lines 50-54: At least here the references must be broader

Recommended literature:

-Usseglio-Polatera P, Richoux P, Bournaud M, Tachet H. 2001. A functional classification of benthic macroinvertebrates based on biological and ecological traits: application to river condition assessment and stream management. Archiv für Hydrobiologie. Supplement-Band 139: 53–83.

-Fabrizi, A., Goretti, E., Compin, A., Céréghino, R., 2010. Influence of fish farming on the spatial patterns and biological traits of river invertebrates in an appenine stream system (Italy). Int. Rev. Hydrobiol. 95 (4-5), 410–427.

-Pallottini M., Goretti E., Gaino E., Selvaggi R., Cappelletti D.,  Cereghino R., 2015. Invertebrate diversity in relation to chemical pollution in an Umbrian stream system (Italy). Comptes Rendus Biologies, 338, 511–520.

Lines 53-54: the references are misplaced, put them at the end of the sentence, all together “….. on macroinvertebrates [4-11]”

Lines 77-78: the first part of the sentence is a repetition of what already stated at lines 43-44

Research area and methods

Lines 84-86: rewrite this sentence, in the present form it is not understandable

Line 99: Table 1

there are some missing information for some of the rivers (i.e., discharge, length and catchment area), complete the table (if possible), at least the length of the Qingyang River! it is not comprehensible what “station number” represents (sampling station or hydropower stations!). The sum of the numbers in the column is 48, the total samples should be 133 ( in Table samples 114: missing for Nan River the number of samples at 2012-Jun, probably 19 samples!). the column “sample number/time” is very messy and not understandable (e.g. “11/201111” to replace with “11/2011-Nov”…).

AFTER LINE 99 THE LINE NUMBERING DISAPPEARS

2.2.1

- “In this study, macroinvertebrates in the rivers were collected using a nylon yarn D-frame net (width: 0.30 m, mesh size: 450 μm)”: the sample surface is not indicated! In fact, in this subsection should be important to understand which was the surface area sampled at each site, this aspect it’s not clear.

- Replace “Peterson graph” with “Peterson grab”

- why only tissues from Mollusca were dried to calculate the biomass?

- In the sentence “Macroinvertebrates were identified to the lowest possible taxon in the laboratory with the aid of a dissecting microscope by [17-18] using identification keys [19]”, the references should be cited at the end of this sentence [17-19]

2.2.3

- replace “samples” with “specimens” in all the subsection

Results and discussion

 3.2.1

- Table 2: widen first column (hydrological in only one line)

- Fig. 3 is a cumulative figure that serves no purpose. It is necessary for each river to report density and biomass at the different stations separating the two hydrological periods. If this documentation is too large for the manuscript it can be reported in the supplementary material.

- Fig. 4 missing caption for reservoir, down-dam, water-reducing reaches, mixing reaches, and natural reach Also report the percentages of variance in the axes to help the reader.

- Fig. 5: how many samples were taken at NR river? It’s not clear, see also Table 1 comment

Also report the percentages of variance in the axes to help the reader.

-The Nan river is taken as an example, so it is necessary at least in material supplementary to indicate the position of the sampling stations with respect to the hydroelectric station in a figure / scheme

- Tabs. 3,4,5: “down-dam reaches” and not only “downdam”, use consistency

- move “(Table 4)” at the end of the sentence

- macroinvertebrates were identified at the lowest taxonomical level possible, use the word “taxa” and not “species”. This note is also valid for the entire text of the Manuscript.

- replace the sentence “The statistical results for the proportion of individual macroinvertebrate species in each phylum in the reservoirs, down-dam reaches, and natural reaches showed in Table 5.” With The statistical results for the proportion of individual macroinvertebrate species in each phylum in the reservoirs, down-dam reaches, and natural reaches ARE showed in Table 5.”

- Table 5: add in the caption that data are referred to NR

- Diptera Chironomidae are not “mosquito”, you can refer to them as “midges” or “non-biting midges”

3.2.2

-The Qingyang river is taken as an example, so it is necessary at least in material supplementary to indicate the position of the sampling stations with respect to the hydroelectric station in a figure / scheme

Conclusion

I would recommend starting the sentence from the sentence: “In summary, the influence of hydropower stations on macroinvertebrate communities exhibited obvious differences between the dam- and diversion-type power stations. For dam-type hydropower stations, a reservoir forms because of the high dam.”  Therefore it is necessary to rewrite this final part  of the text

References

It should be enriched

Supplementary Material

Insert in this section the macroinvertebrate tables for the 7 rivers in the different stations

Round 2

Reviewer 2 Report

Please see my review comments in the attachment provided. 

Reviewer 3 Report

Only minor corrections

-Line 109: Table 1

the total samples should be 133 (in the Table the samples are126!, for Nan River the number of samples at 2012-Jun, probably are 19 samples)

-Line 152: Figure 2 Caption 

Replace “Figure 2. Species composition of macroinvertebrates (A) and aquatic insects (B).” With “Figure 2. Taxa composition of macroinvertebrates (A) and aquatic insects (B).”
